# Patterns of Use of Vaping Products among Smokers: Findings from the 2016–2018 International Tobacco Control (ITC) New Zealand Surveys

**DOI:** 10.3390/ijerph17186629

**Published:** 2020-09-11

**Authors:** Richard Edwards, James Stanley, Andrew M. Waa, Maddie White, Susan C. Kaai, Janine Ouimet, Anne C.K. Quah, Geoffrey T. Fong

**Affiliations:** 1Department of Public Health, University of Otago, Wellington 6023, New Zealand; james.stanley@otago.ac.nz (J.S.); andrew.waa@otago.ac.nz (A.M.W.); maddie.white@uni-bremen.de (M.W.); 2Department of Psychology, University of Waterloo, 200 University Ave W., Waterloo, ON N2L 3G1, Canada; skaai@uwaterloo.ca (S.C.K.); j2ouimet@uwaterloo.ca (J.O.); ackquah@uwaterloo.ca (A.C.K.Q.); geoffrey.fong@uwaterloo.ca (G.T.F.); 3School of Public Health and Health Systems, University of Waterloo, 200 University Ave W., Waterloo, ON N2L 3G1, Canada; 4Ontario Institute for Cancer Research, 661 University Ave Suite 510, Toronto, ON M5G 0A3, Canada

**Keywords:** E-cigarettes, vaping, prevalence, knowledge, risk perceptions, policy

## Abstract

Alternative nicotine products like e-cigarettes could help achieve an end to the epidemic of ill health and death caused by smoking. However, in-depth information about their use is often limited. Our study investigated patterns of use of e-cigarettes and attitudes and beliefs among smokers and ex-smokers in New Zealand (NZ), a country with an ‘endgame’ goal for smoked tobacco. Data came from smokers and ex-smokers in Waves 1 and 2 of the International Tobacco Control (ITC) NZ Survey (Wave 1 August 2016–April 2017, 1155 participants; Wave 2, June–December 2018, 1020 participants). Trial, current and daily use of e-cigarettes was common: daily use was 7.9% among smokers and 22.6% among ex-smokers in Wave 2, and increased between surveys. Use was commonest among 18–24 years and ex-smokers, but was similar among Māori and non-Māori participants, and by socio-economic status. Most participants used e-cigarettes to help them quit or reduce their smoking. The most common motivating factor for use was cost and the most common barrier to use cited was that e-cigarettes were less satisfying than smoking. The findings could inform developing interventions in order to maximise the contribution of e-cigarettes to achieving an equitable smoke-free Aotearoa, and to minimise any potential adverse impacts.

## 1. Introduction

Research into e-cigarettes or vaping products and other alternative nicotine delivery devices (ANDs) is an increasing focus in tobacco control research and there is ongoing discussion about the possible contribution (positive or negative) of vaping to the achievement of smoke-free goals. Hence, there is interest in patterns of use, beliefs and understandings about vaping products in jurisdictions with different levels of development, tobacco use prevalence, and policy and regulatory environments. However, in-depth information on these topics is limited in many countries.

Research on e-cigarettes and vaping in New Zealand is relevant for several reasons. First, in New Zealand, like many other countries, use of e-cigarettes is increasing. For example, prevalence of adult daily use increased almost fourfold from 0.9% in 2015–2016 to 3.2% in 2018–2019 [1]. Second, New Zealand has a tobacco control context with several features of interest. It has implemented most of the core tobacco control measures of the WHO Framework Convention on Tobacco Control (FCTC) and is one of only a few countries with a Government endorsed ‘endgame’ goal—to reduce smoking prevalence and tobacco availability to minimal levels by 2025 [2]. Third, New Zealand has a relatively low and declining smoking prevalence (current smoking prevalence 14.2% in 2018–2019), but persisting disparities in smoking with particularly high prevalence among Māori—the indigenous peoples of New Zealand (current adult smoking prevalence 34.0% in 2018–2019) [1].

Finally, there is an interesting regulatory context for vaping and other ANDs in New Zealand. Nicotine-containing e-cigarettes were initially prohibited for sale in New Zealand and were only legally available through personal importation via online sales. However, following a Court judgement in March 2018 determining that heated tobacco products could be legally sold in New Zealand, there was a rapid and largely unregulated increase in the availability and marketing of nicotine-containing vaping products. After March 2018 vaping products were initially sold mainly in specialist stores, but are now increasingly available in non-specialist retailers such as convenience stores and gas stations. A wide range of devices and e-liquids are available, with no regulation over constituents, flavours or nicotine content and marketing is increasingly widespread. Hence, there is active debate about the appropriate regulatory framework, and new legislation, the Smoke-free Environments and Regulated Products (Vaping) Amendment Bill [3,4], regulating the availability, constituents and marketing of vaping products and other ANDs was approved by Parliament in August 2020.

Despite this context, there is limited population level information on vaping product use and related attitudes and beliefs in New Zealand, and whether these have changed over time. The main nationally-representative adult surveys are the New Zealand Health Survey and the Health Promotion Agency’s Health and Lifestyle surveys. These provide high quality evidence about prevalence of use [1,5] but do not collect detailed information on patterns of use or beliefs and attitudes. An online survey from 2016 provided such information, but was restricted to a small sample of established vapers (*n* = 218) recruited mainly through specialist vaping retailers, vaper groups and social media, and hence was unlikely to be representative of all vapers. This survey also included only a small number of people who smoked at the time of the survey (*n =* 46), most of whom were occasional smokers, so provided only limited information on vapers who are currently smoking [6].

This paper provides detailed information about patterns of use and knowledge, attitudes and beliefs about e-cigarettes among a nationally-representative sample of smokers and ex-smokers in New Zealand, with findings also described for Māori and non-Māori participants and by smoking status. The aim is to provide information to inform the development of regulations and legislation and to provide evidence about the possible contribution of vaping to reducing smoking and disparities in smoking, and hence to achieving New Zealand’s smoke-free goal.

## 2. Materials and Methods

### 2.1. Study Population and Sampling

Data came from smokers and ex-smokers included in the first and second waves of the International Tobacco Control New Zealand (ITC NZ) Survey. Participants for Wave 1 were recruited from a sampling frame of smokers and ex-smokers who took part in the NZ Health Survey (NZHS) between 1 January 2015 and 30 June 2016 and who had agreed to be contacted about further research projects. The NZHS is a nationally representative survey of NZ adults. Wave 2 participants included Wave 1 respondents who agreed to participate in a follow-up survey and replenishment participants recruited from smokers and ex-smokers who had recently quit, amongst NZHS respondents from 1 July 2016 to 30 June 2018.

Eligible participants were defined as current smokers (smoked more than 100 cigarettes in their lifetime and currently smoked at least monthly) or ex-smokers (smokers who had quit smoking in the previous year at the time of their NZHS interview) aged ≥ 18 years. The sampling scheme aimed to over-sample four priority sub-groups to increase analysis precision for these groups: (1) Māori current smokers, (2) Pacific current smokers, (3) younger current smokers (non-Māori, non-Pacific smokers aged 18–24), and (4) and ex-smokers (quit within the year prior to their NZHS interview).

### 2.2. Data Collection

We collected data through computer-aided telephone interview surveys conducted from August 2016 to April 2017 (Wave 1) and June to December 2018 (Wave 2). The mean duration of the interviews was 77 min for smokers (both waves) and 69 min (Wave 1) and 67 min (Wave 2) for ex-smokers.

Full details of the sampling and survey methods are available in the Wave 1 and Wave 2 ITC New Zealand Technical Reports available online [7,8].

### 2.3. Measures

The first question about awareness of e-cigarettes described them as: ‘electronic cigarettes, also called e-cigarettes, vapes or vaping devices’. Subsequent questions referred to ‘e-cigarettes or vaping devices’. For brevity, this manuscript will henceforth refer to e-cigarettes or vaping, as appropriate.

Participants were asked whether they were aware of and whether they had ever used e-cigarettes. Current users were asked about their frequency of use. From the responses to these questions, we classified participants as daily users, weekly/monthly users, occasional (less than monthly), or ex-users and never users. Any participant who used e-cigarettes at least monthly was classified as a current user. We also asked current users about the type of e-cigarette device that they used, whether they used nicotine-containing e-cigarettes or e-liquids, which flavour(s) they most commonly used, and where they had made their most recent purchase of an e-cigarette, cartridge or e-liquid. Finally, we asked current users about the reasons and motivations for using e-cigarettes, and we asked all participants who were aware of e-cigarettes about their beliefs about the health effects and addictiveness of e-cigarettes and of some potential barriers to their use. The Appendix A includes the questions and response options. The full questionnaire can be accessed online [9].

### 2.4. Analysis

We present cross-sectional analyses from Wave 1 and 2 of the ITC NZ survey. Due to the stratified sampling method with differential recruitment by age and ethnicity and the complex survey nature of the NZHS sampling frame, all reported analyses use inverse sample weights so that estimates apply to the relevant total NZ populations of smokers and recent quitters (henceforth referred to as ex-smokers, as recent quit status was at the time of the original NZHS interview rather than at the time of the ITC interview). Weights were calibrated based on the NZHS data for these populations: see Appendix E of the ITC NZL Wave 1 report and Appendix F of the ITC NZL Wave 2 report for full details [7,8].

We present descriptive statistics of responses (prevalence/percentages) among all participants, and stratified as appropriate by age group (18–24 vs. 25+), gender (male and female), self-identified ethnicity (Māori and non-Māori, with ethnicity prioritised so that all participants who identified as Māori and one or more other ethnicity were classified as Māori), socio-economic status (quintiles of NZ Dep, an area based measure of deprivation) [10], smoking status (smokers and ex-smokers), and frequency and history of vaping (current, daily, weekly, or monthly, less than daily or ex-vaper, never vaper). For some analyses, smokers were also split between those that had and hadn’t made a quit attempt in the previous year.

Prevalence trends are presented with data from Waves 1 and 2, whilst detailed patterns of use and knowledge, beliefs, and attitudes about vaping products are presented from Wave 2. Participants are included in all analyses for which they had responses. Unless otherwise stated participants who responded ‘don’t know’ or refused to respond have been excluded from the analyses presented. The exceptions were questions on beliefs about vaping and vaping products, where ‘don’t know’ responses were treated as valid answers.

Percentages for sub-populations were calculated using marginal standardisation to adjust for differences in sociodemographic and smoking characteristics between these groups (age group, gender, Māori and non-Māori, and where appropriate smoking status as smoker/quitter). Marginal standardisation is akin to direct age standardisation (as commonly used in epidemiology when comparing rates by groups) but usually involves adjustment for a wider range of covariates, and with estimation based on an initial logistic regression model [11]. Sub-group estimates for Wave 2 included further adjustment for whether a respondent had taken part in both waves or was a replenishment participant recruited for Wave 2 (binary variable). Results are reported with 95% confidence intervals (95% CI).

We conducted logistic regression analyses adjusting for the variables used in marginal standardisation and additional potential confounders (NZDep quintile and more detailed smoking status that included whether current smokers had made a quit attempt in the last year) to investigate for independent associations between sociodemographic factors and smoking status and e-cigarette use.

We present prevalence data with indicators of precision (95% confidence limits) on reasons and motivators for use and beliefs and attitudes about e-cigarettes, stratified by smoking status and history and frequency of e-cigarette use. In making comparisons between sub-populations, we have not carried out statistical significance tests because such tests risk focusing the evaluation of the importance and validity of findings on an arbitrary level of statistical significance, and would be difficult to interpret given the high risk of type 1 error from multiple significance testing. We have instead drawn attention to findings that we think are noteworthy due to the size of the observed differences or consistency in the pattern of the result across sub-populations. This approach is in line with current statistical thinking rejecting over-reliance on tests of statistical significance [12,13].

All analyses were conducted in R 4.0.1 (R Institute, Vienna, Austria) using the ‘survey’ package to account for the complex survey design.

### 2.5. Ethical Approval

Ethics review and approval were given by the University of Otago Human Ethics Committee (application number 15/126) and University of Waterloo Office of Research Ethics (ORE 30726). 

## 3. Results

### 3.1. Sample Characteristics and Response

The Wave 1 sample of 1155 participants was made up of 910 smokers and 245 ex-smokers and included 386 identifying as Māori. The Wave 2 sample of 1020 participants included 726 smokers and 294 ex-smokers, including 394 identifying as Māori. The Wave 2 sample comprised 587 returning participants (57.5%) and 433 replenishment participants (42.5%). Overall response (the proportion of all invited potential participants from the NZHS sampling frame who took part in the survey) in Wave 1 was 27.6% and in Wave 2 was 19.8% Consent to participate (the proportion of all invited potential participants from the NZHS sampling frame who were successfully contacted and who took part in the survey) was 41.5% in Wave 1 and 32.5% in Wave 2 [7,8]. Table 1 shows the characteristics of the Wave 1 and Wave 2 samples. Not all participants answered all questions: denominators for individual questions are given as appropriate in the tables.

### 3.2. Prevalence and Patterns of Use of Vaping Products

Table 2 shows prevalence of awareness and use of e-cigarettes; comparing Wave 1 and Wave 2 illustrates recent shifts in patterns of use. Appendix A provides estimates of the absolute differences in these measures adjusted for differences in sample composition between Wave 1 and Wave 2.

Awareness was almost universal (98.0% Wave 2). There was a substantial increase in ever use between Wave 1 (59.4%) and Wave 2 (76.9%), a 17.2% absolute increase; (95% CI 12.7–21.8: see Appendix A). Current (at least monthly) use and daily use of e-cigarettes also increased from Wave 1 (16.1% and 8.0% respectively) to Wave 2 (22.2% and 11.4%) among all participants (absolute differences by wave in Appendix A).

Current use among ex-smokers was substantially higher than among smokers. For example, in Wave 2, at least monthly use was 28.8% for ex-smokers vs. 20.0% for smokers (absolute difference: 8.2%, 95% CI 1.3–16.4, *p* = 0.022) whilst differences in daily use were more substantial (22.6% vs. 7.9% respectively); absolute difference 14.7%, 95% CI 7.8–21.6, *p* < 0.001). Differences in ever use were less striking and in the opposite direction, with smokers having higher prevalence of ever use than ex-smokers; absolute difference 5.3% (95% CI—1.5–12.2, *p* = 0.123).

Table 3 shows patterns of ever, current, and daily use of e-cigarettes stratified by gender, age group, ethnicity, socio-economic status, and smoking status in Wave 2. At least monthly and daily e-cigarette use were more common among males, but ever use was similar for males and females. Current and daily use were similar between Māori and non-Māori, but Māori participants were more likely to have ever tried e-cigarettes. Differences in e-cigarette use by quintiles of deprivation were mostly small and non-statistically significant from the reference group. The only exception was higher ever use of e-cigarettes among participants in NZDep quintile 4 (2nd most deprived quintile) compared to the most deprived quintile. Prevalence of use (all measures) was markedly higher among young adults (18–24 years) than older participants, and current and daily use was higher among ex-smokers and smokers who had tried to quit in the last year compared to smokers who had not.

### 3.3. Type of Products Used and Place of Purchase

In Wave 2, among current (at least monthly) users of e-cigarettes (*n* = 204), 86.4% used third generation (rechargeable with a refillable tank) devices and 13.5% second generation pre-filled devices; 80.8% were currently using a nicotine-containing e-cigarette or e-liquid. The number of flavours used by current users was approximately evenly split between one flavour (37.3%), two flavours (29.5%), and three or more flavours (33.2%). The most commonly used flavours among current users (i.e., most commonly used flavour reported per person) were: fruit (39.4%), tobacco (23.8%), candy/sweet/dessert (16.6%), and menthol/mint (11.3%).

Wave 2 current users were most likely to have last purchased e-cigarettes, e-liquid or cartridges at a vape shop (55.9%), followed by the internet (15.4%), tobacconist (9.7%), and a local convenience store (5.4%).

### 3.4. Reasons and Motivators for Using E-Cigarettes

Current users (at least monthly) were asked about their reasons for using e-cigarettes or vaping. The most frequent responses are shown in Table 4 divided into reasons for use such as to quit smoking and motivating factors such as e-cigarettes being cheaper than smoking.

The column ‘*n*’ is the number of participants in the column group with at least one valid answer to this question set, and the row *‘n*’ is the total number of participants with valid responses to that specific question.

Most smokers reported that they use e-cigarettes to help them quit (74.7%) or to cut down on their smoking (86.7%). However, a substantial proportion of smokers (60.7%) somewhat contradictorily said they vaped so that they didn’t need to quit smoking completely; and 40.3% stated that a reason for using e-cigarettes was that they could vape in areas where smoking wasn’t allowed. Ex-smokers who used e-cigarettes were more likely (86.7%) to state that the reason they used e-cigarettes was to help them quit smoking.

Over half of participants agreed with each of the five potential motivators for use, with findings broadly similar between smokers and ex-smokers. The most commonly cited motivators were that e-cigarettes are cheaper than smoking (90.6%) and enjoyment of vaping (72.2%). There were marked differences in health-related beliefs between smokers who had made a quit attempt in the previous year and those who had not. The former were much more likely to believe e-cigarettes were less harmful than cigarettes both to the users and to others exposed to second-hand vapour.

### 3.5. Attitudes and Beliefs about E-Cigarettes and Vaping

Beliefs about potential barriers to use, harmfulness and addictiveness of e-cigarettes among participants who were aware of e-cigarettes are shown in Table 5, stratified by frequency and history of e-cigarette use and by smoking status. Responses were mostly similar between smokers and ex-smokers, but there were some substantial differences in relation to e-cigarette use.

A small proportion of participants agreed that e-cigarettes were too hard to get (9.5%) or too complicated to use (22%). Responses were generally similar for sub-groups (considered by frequency and history of e-cigarette use), except never users were more likely to be unsure whether e-cigarettes were too hard to get or too complicated to use.

Most participants thought vaping with nicotine e-cigarettes was less (36.5%) or equally (43.1%) addictive as smoking cigarettes. Daily users (59.4%) and weekly/monthly users (57.8%) were more likely to believe that e-cigarettes were less addictive than occasional/ex-users (33.2%), or never users (19.4%). Over two-thirds (68.3%) of participants believed e-cigarettes to be less satisfying than smoking cigarettes, including about three-quarters (76.9%) of weekly/monthly users and 70.5% of occasional and ex-users. This belief was less common among daily users, but was still present for over half (53.3%) of this group. This question was not asked of never users.

Most (62.4%) participants thought vaping with e-cigarettes was cheaper than smoking cigarettes, including almost all daily users (94.7%). However, of never users only 29.4% expressed this belief and almost half (43.3%) were unsure. Around half of all participants (46.2%) thought e-cigarettes were less harmful than smoking cigarettes, with most of the remainder (33.0%) thinking that vaping was equally harmful. However, just over three-quarters (76.0%) of daily users and 68.2% of weekly/monthly users thought vaping was less harmful than smoking, whilst this belief was much less common among occasional/ex-users (43.6%) and never users (24.5%).

For the six questions about beliefs, ‘don’t know’ responses varied between 9.5% and 21.5%: the proportion of respondents answering ‘don’t know’ for a given question was lowest for daily (0%–3.1%) and weekly/monthly (0.9%–9.9%) users, higher for occasional/ex-users (8.7% to 19.8%), and highest for never users (19.8%–43.3%).

## 4. Discussion

The results provide a comprehensive picture of e-cigarette use and attitudes and beliefs about e-cigarettes among smokers and ex-smokers in New Zealand. Trial and use of e-cigarettes were common among New Zealand smokers and quitters, and increased from 2016–17 to 2018. In a recent cross-country analysis of current smokers in 14 ITC countries, the prevalence of daily e-cigarette use in NZ was second only to England [14].

Trial and regular use was most common among 18–24 year olds, but was otherwise broadly similar among Māori and non-Māori and by gender and socio-economic status. However, current and daily use of e-cigarettes was substantially more prevalent among ex-smokers. A high prevalence of daily use among ex-smokers (higher than current smokers) was also found in an analysis of national survey data [5]. Prevalence of daily use among ex-smokers and the differences in daily use prevalence between ex- and current smokers were highest in New Zealand in ITC analyses across 14 countries [14]. These findings, together with the high proportion of users who said they vaped as an aid to quit smoking, suggests that e-cigarettes may be supporting quitting among a significant number of people who smoke in New Zealand. On the other hand, it was concerning that over half of smokers stated they used e-cigarettes to replace some of their cigarettes as an alternative to quitting smoking completely, as this could result in use of e-cigarettes reducing the likelihood of quitting among some of these smokers.

We found prevalence of ever, current and daily use were mostly similar among Māori and non-Māori participants (other than moderately higher ever use among Māori), and also by socio-economic status (as assessed by an area-based measure of deprivation). Given the unclear overall pattern of prevalence by NZDep quintile, the higher prevalence of ever use in NZDep quintile 4 may be a random chance variation. These findings suggest that currently the impact of e-cigarettes will be broadly neutral and will neither exacerbate nor decrease disparities in smoking in New Zealand.

Many of these findings are relevant to policy and practice, particularly important as the New Zealand Parliament recently passed new vaping regulation legislation (August 2020) [3,4]. For example, the findings that e-cigarette use was most prevalent among 18–24 year old smokers/quitters, suggests that efforts to promote the use of vaping as a cessation tool or substitute for smoking need to focus on older age groups. It is concerning, therefore, that recent reports suggest that vaping products produced by the tobacco industry have been marketed using approaches which appear to target youth and young adults [15]. The high prevalence of use of flavours and the wide range of flavours used by participants aligns with the findings of other studies and suggests that if a policy aim is to promote smokers to use e-cigarettes to quit or to switch to vaping, making vaping products available in multiple flavours may encourage their use by smokers [16]. However, there is also evidence of increasing vaping among New Zealand adolescents [17] and there are concerns that some flavours may promote youth uptake, so a regulatory balance needs to be struck in order to minimize the risk of youth use of e-cigarettes [18]. The new New Zealand vaping regulation legislation allows all flavours of e-liquids and cartridges to be sold by licensed specialist vaping shops, but only three flavours (tobacco, mint and menthol) by non-specialist stores such as convenience stores and gas stations. There is active debate in New Zealand as to whether this provides an appropriate regulatory balance.

At the time of Wave 2 of the survey in 2018 e-cigarettes were not yet widely available for purchase in New Zealand outside of specialist vape stores. This was reflected in the finding that participants most commonly reporting buying e-cigarettes from specialist stores. The finding that only a minority of participants thought e-cigarettes were too hard to get suggests specialist stores provided adequate access to vaping products for most smokers. However, that may not be true for smokers in more remote areas without access to a specialist store. Constraining access to these products at other stores such as convenience stores and petrol stations has been argued to have benefits. These include minimising youth access and maximising access to expert advice about vaping products and their use for smokers who are starting to vape [19], though there are also contrary views [20].

There have been regular above inflation tobacco excise tax increases since 2010 with the result that cigarettes by 2018 cost around NZ $35 (equivalent to US $23) per pack of 25, and were among the most expensive cigarettes in the world. According to the calculator on the New Zealand Government’s ‘Vaping Facts’ website, a typical 10 cigarette per day smoker in New Zealand would spend NZ $4909 per year on smoking compared to NZ $394 for a typical vaper [21]. The most commonly cited motivation for use of e-cigarettes in our study was cost. This suggests that the current New Zealand situation with very high taxes on smoked tobacco products, making them much more expensive than e-cigarettes, provides an appropriate stimulus for smokers to use e-cigarettes to quit or as substitutes for smoking, if that is the policy aim. This finding illustrates the benefits of proportionate regulation of nicotine products according to their degree of harmfulness and the potential synergy between robust policies to discourage smoked tobacco use on the one hand and less stringent policies for the regulation of vaping products [22,23]. However, the need for introducing specific excise taxes to influence e-cigarette use may need to be reconsidered if use among New Zealand adolescents continues to increase [17,18]. In addition, the findings suggest that raising awareness among smokers about the relative costs of vaping vis-a-vis smoking may be helpful, as many non-users did not know that vaping is substantially cheaper than smoking.

Another important motivating factor for e-cigarette use was lower perceived harmfulness of vaping compared to smoking. Almost half of participants and a large majority of daily users perceived e-cigarettes to be less harmful than smoking. However, a substantial proportion of participants, particularly non-users, thought vaping was more or equally harmful or were unsure about the relative harmfulness of smoking and vaping, suggesting that further education of smokers about the relative harmfulness of these products would be useful if their use among smokers is a desired aim. The recently launched Government ‘Vaping Facts’ website [24] may help in this regard.

The most common factor discouraging smokers from using e-cigarettes was the belief of a high proportion that vaping is less ‘satisfying’ than smoking. This was particularly common among non-daily users. This suggests that either e-cigarettes are inadequate substitutes for smoking for most smokers (although the higher level of satisfaction with vaping among daily users suggests that they can be acceptable substitutes for many smokers), or that many vapers are not using the most appropriate device. If the second proposition is true, and policy-makers want to encourage e-cigarette use by smokers, it provides another argument for enhancing public education to ensure smokers are better informed about vaping, and to implement regulatory approaches which encourage or require users to source e-cigarettes in places where more expert advice is available (e.g., specialist vape stores) about the best device and e-liquids to use [25].

Strengths of this study include the use of a nationally representative survey as a sampling frame (enhancing generalisability), a wide range of relevant questions, and availability of repeat cross-sectional data allowing the exploration of trends. Limitations include the self-reported nature of the data, with the risk of social desirability bias, and the relatively low response rate with the risk of selection bias. In addition, because the sample was restricted to adult smokers and ex-smokers, the study cannot provide direct evidence about the impact of vaping on population smoking prevalence or other important issues such as vaping and smoking among youth (under the age of 18).

Further analysis of the ITC data will enable investigation of how attitudes and beliefs to e-cigarettes and vaping vary by ethnicity and socio-economic status. Although the ITC NZ wave 2 findings provide the most detailed population-based data on vaping in New Zealand, patterns of use were described only at the start of the period of rapid change in the e-cigarette market that followed the March 2018 court ruling. For example, subsequently e-cigarettes have become available in a much wider range of stores, marketing of vaping products has increased greatly, the multi-national tobacco industry has introduced its products into the New Zealand market (previously the New Zealand market was dominated by independent vaping companies), and new technologies like ‘pod’ devices have become available. Further quantitative research is therefore needed to investigate how patterns of use of vaping and related products and their impacts on smoking have evolved in 2019 and 2020, and also to explore the impact of the new regulatory and legislative framework that will be introduced following the passing into law of the vaping amendment Bill. Such studies are particularly important with people from high smoking prevalence groups and among smokers who have not tried or are unwilling to use e-cigarettes.

## 5. Conclusions

The use of e-cigarettes among smokers and ex-smokers in New Zealand is increasing in prevalence, with most smokers and ex-smokers using e-cigarettes to help them quit or to reduce their smoking. Prevalence of use was similar for Māori and non-Māori participants and varied little by socio-economic status. The findings concerning reasons for use and beliefs about vaping could inform the development of evidence-based policy and interventions to maximise the impacts of vaping in supporting the achievement of an equitable Smoke-free Aotearoa, and to minimise any adverse impacts of these products such as uptake by young people and non-smokers.

## Figures and Tables

**Table 1 ijerph-17-06629-t001:** Sample characteristics in Waves 1 and 2 of the International Tobacco Control (ITC) New Zealand Survey.

Characteristic	Wave 1 (*n* = 1155) *n* (Unweighted %)	Wave 2 (*n* = 1020) *n* (Unweighted %)
Age		
18–24 years	93 (8.1)	87 (8.5)
≥25 years	1062 (91.9)	933 (91.5)
Gender		
Female	671 (58.1)	622 (61.0)
Male	484 (41.9)	398 (39.0)
Ethnicity		
Māori	386 (33.4)	394 (38.6)
Non-Māori	769 (66.6)	626 (61.4)
NZDep 2013 Quintile		
1 (least deprived)	92 (8.0)	74 (7.3)
2	132 (11.4)	116 (11.4)
3	199 (17.2)	164 (16.1)
4	312 (27.0)	283 (27.7)
5 (most deprived)	420 (36.4)	383 (37.5)
Smoking status		
Current smoker (quit attempt in last year)	487 (42.2)	380 (37.2)
Current smoker (no quit attempt in last year)	423 (36.6)	346 (33.9)
Ex-smoker	245 (21.2)	294 (28.8)

**Table 2 ijerph-17-06629-t002:** Prevalence of awareness and use of e-cigarettes in Waves 1 and 2 stratified by smoking status.

Measure	Wave 1 ^1^			Wave 2 ^2^		
	Total	Smokers	Ex-smokers	Total	Smokers	Ex-smokers
	% (95% CI)	% (95% CI)	% (95% CI)	% (95% CI)	% (95% CI)	% (95% CI)
	(*n* = 1086)	(*n* = 857)	(*n* = 229)	(*n* = 1020)	(*n* = 726)	(*n* = 294)
Awareness of vaping products ^3^	93.7	93.5	94.6	98.0	98.0	98.0
	(91.9–95.2)	(91.4–95.2)	(90.1–97.1)	(96.5–98.9)	(96.4–98.9)	(95.1–99.2)
Ever used vaping products ^3^	59.3	59.7	57.9	76.9	78.2	72.8
	(55.7–63.1)	(55.5–63.9)	(50.4–65.7)	(73.6–79.9)	(74.4–81.5)	(66.4–78.3)
At least monthly current use of vaping products ^4^	16.1	14.2	23.8	22.2	20.0	28.8
	(13.3–19.3)	(11.4–17.5)	(16.8–32.5)	(18.8–26.0)	(16.2–24.4)	(22.8–35.8)
Daily current use of vaping products ^4^	8.0	4.9	21.0	11.4	7.9	22.60
	(6.0–10.6)	(3.3—7.2)	(14.6–30.3)	(9.1–14.1)	(5.6–11.1)	(17.0–29.5)

^1^ Estimates for smokers and quitters are marginally standardised on age, gender and ethnicity; ^2^ Wave 2 estimates for smokers and quitters are further standardised for time in sample; ^3^ Valid answers at Wave 1: *n* = 1086; Wave 2: *n* = 1020; ^4^ Valid answers at Wave 1: *n* = 1039; Wave 2: *n* = 1012.

**Table 3 ijerph-17-06629-t003:** Trial and use of e-cigarettes in Wave 2: stratified estimates and associations with gender, ethnicity, age group, socio-economic status, and smoking status.

Characteristic	Ever Use % (95% CI) *n* = 1020	AOR ^3^ (95% CI) *n* = 1020	Current Use ^1^ % (95% CI) *n* = 1012	AOR ^3^ (95% CI) *n* = 1012	Daily Use ^2^ % (95% CI) *n* = 1012	AOR ^3^ (95% CI) *n* = 1012
All	76.9 (73.6–79.9)		22.2 (18.8, 26.0)		11.4 (9.1, 14.1)	
Gender						
Female	77.3 (73.2–81.1)	1.00 (Reference)	18.2 (14.7–22.5)	1.00 (Reference)	10.0 (7.5–13.2)	1.00 (Reference)
Male	76.4 (71.4–80.8)	0.97 (0.67–1.41)	26.8 (21.2–33.3)	1.89 (1.24–2.90)	13.0 (9.1–18.2)	1.51 (0.86–2.64)
Ethnicity						
Non-Māori	75.1 (71.0–78.8)	1.00 (Reference)	21.7 (17.8–26.3)	1.00 (Reference)	11.5 (8.6–15.1)	1.00 (Reference)
Māori	81.6 (76.6–85.8)	1.58 (1.06–2.38)	23.4 (17.5–30.7)	1.15 (0.71–1.86)	11.1 (7.7–15.7)	1.05 (0.59–1.81)
Age group						
≥45 years	64.9 (60.0–69.5)	1.00 (Reference)	17.0 (13.7–20.8)	1.00 (Reference)	10.4 (7.9–13.5)	1.00 (Reference)
25–44 years	82.0 (76.6–86.4)	2.46 (1.66–3.69)	21.1 (16.2–27.1)	1.26 (0.81–1.97)	10.0 (6.7–14.5)	0.93 (0.54–1.61)
18–24 years	93.7 (86.8–97.1)	8.37 (3.76–21.64)	41.7 (28.7–56.0)	3.69 (1.88–7.37)	18.6 (10.1–31.7)	1.97 (0.84–4.56)
NZ Dep Quintile						
5 (most deprived)	73.6 (67.9–78.7)	1.00 (Reference)	24.1 (18.9–30.2)	1.00 (Reference)	10.7 (7.5–15.1)	1.00 (Reference)
4	82.2 (77.1–86.4)	1.72 (1.10–2.71)	29.7 (22.7–37.8)	1.39 (0.84–2.32)	12.2 (7.9–18.3)	1.20 (0.62–2.31)
3	73.6 (65.2–80.7)	1.01 (0.59–1.72)	15.3 (10.0–22.8)	0.57 (0.30–1.06)	11.4 (7.1–17.9)	1.13 (0.55–2.29)
2	78.8 (69.9–85.6)	1.37 (0.78–2.48)	18.4 (10.8–29.5)	0.70 (0.33–1.45)	12.8 (6.6–23.4)	1.27 (0.53–2.91)
1 (least deprived)	77.2 (64.4–86.3)	1.25 (0.61–2.68)	19.3 (11.0–31.7)	0.84 (0.38–1.82)	9.8 (4.9–18.6)	1.08 (0.42–2.62)
Smoking status						
Smoker (no recent quit attempt)	76.9 (72.0–81.1)	1.00 (Reference)	15.3 (10.6–21.6)	1.00 (Reference)	4.3 (2.2–8.4)	1.00 (Reference)
Smoker (recent quit attempt)	79.7 (73.8–84.5)	1.20 (0.77–1.87)	25.6 (20.0–32.1)	1.93 (1.14–3.32)	12.3 (8.3–17.7)	3.11 (1.36–7.69)
Ex-smoker	72.8 (66.5–78.3)	0.76 (0.50–1.16)	29.1 (23.0–36.0)	2.31 (1.32–4.09)	22.9 (17.2–29.8)	6.66 (2.95–16.35)

Note: Prevalence estimates for sub-groups are marginally standardised for age group, gender, ethnicity, smoking status (current/ex-smoker), time in sample; ^1^ Current use—at least monthly e-cigarette use (daily/weekly/monthly) vs. non-current use (less than monthly, past or never use); ^2^ Daily use—daily e-cigarette use vs. less frequent (weekly/month use) or non-current use; ^3^ Adjusted Odds ratio—mutually adjusted for age group, gender, ethnicity, NZDep, smoking status (current smoker with no quit attempt last year/current smoker with quit attempt last year/ex-smoker) and time in sample.

**Table 4 ijerph-17-06629-t004:** Most commonly cited reasons and motivators for e-cigarette use among current (at least monthly) e-cigarette users in Wave 2 stratified by smoking status.

Reason or Motivator for Use	Total % (95% CI) (*n* = 197)	Smokers % (95% CI) (*n* = 119)	Smokers (Quit Attempt in Last Year) % (95% CI) (*n* = 75)	Smokers (No Quit Attempt in Last Year) % (95% CI) (*n* = 44)	Ex-Smokers % (95% CI) (*n* = 78)
**Reasons for use**					
Help to quit smoking (*n* = 196)	78.1 (67.1–86.2)	74.7 (61.4–84.6)	91.4 (81.1–96.3)	56.9 (38.8–73.3)	86.7 (73.5–93.9)
Help to cut down on smoking (*n* = 193)	81.3 (71.5–88.3)	86.7 (75.8–93.1)	96.2 (90.6–98.6)	75.6 (59.3–86.8)	67.3 (53.4–78.7)
To replace some cigarettes as an alternative to quitting (*n* = 116)		60.7 (47.2–72.7)	54.2 (38.2–69.3)	68.1 (45.8–84.3)	*N*/A
Can use e-cigarettes in smoke-free areas (*n* = 191)	31.4 (22.7–41.7)	40.3 (28.6–53.1)	36.7 (22.6–53.5)	44.1 (26.7–63.1)	12.3 (6.3–22.5)
**Motivators for use**					
Save money by using e-cigarettes instead of smoking (*n* = 194)	90.6 (83.8–94.7)	88.7 (79.9–94.0)	89.7 (75.4–96.1)	87.6 (73.1–94.9)	94.2 (80.8–98.4)
E-cigarettes more acceptable than smoking to people around you (*n* = 188)	64.2 (53.5–73.7)	65.7 (52.6–76.7)	72.2 (57.2–83.5)	58.1 (37.9–76.0)	60.9 (46.7–73.6)
Enjoy using e-cigarettes (*n* = 195)	72.2 (63.0–79.8)	70.1 (59.5–78.9)	75.5 (62.4–85.0)	64.4 (48.7–77.5)	76.9 (63.6–86.3)
E-cigarettes may be less bad for health (*n* = 176)	59.7 (49.0–69.5)	57.3 (43.7–69.9)	78.2 (61.3–89.0)	34.4 (19.8–52.9)	64.6 (50.1–76.8)
E-cigarettes less harmful to other people around you than smoking (*n* = 178)	62.3 (51.6–71.9)	59.7 (46.5–71.7)	74.8 (60.0–85.4)	44.9 (26.8–64.4)	67.9 (53.8–79.4)

Prevalence estimates for sub-groups are marginally standardised for age group, ethnicity, gender and time in sample. N/A = not applicable.

**Table 5 ijerph-17-06629-t005:** Attitudes and beliefs about e-cigarettes among Wave 2 participants aware of e-cigarettes, stratified by history and frequency of e-cigarette use and smoking status.

Attitude or Belief about E-Cigarettes	Response Options	All Participants % (95% CI) *n* = 878	Daily EC Users % (95% CI) *n* = 122	Weekly/Monthly Users % (95% CI) *n* = 89	Occasional and Ex-E-Cigarette Users ^1^ % (95% CI) *n* = 457	Never Users % (95% CI) *n* = 202	Smokers % (95% CI) *n* = 609	Ex-smokers % (95% CI) *n* = 269
**Beliefs about potential barriers to vaping**								
E-cigarettes are too hard to get (*n* = 864)	Agree	9.5 (7.4–12.2)	11.5 (5.8–21.3)	7.9 (3.0–19.3)	8.3 (5.7–11.7)	12.3 (7.5–19.4)	10.3 (7.6–14.0)	7.5 (4.8–11.6)
	Neither agree nor disagree	4.0 (2.3–7.0)	0.3 (0.0–2.1)	1.4 (0.4–5.4)	5.3 (2.4–11.3)	4.1 (1.7–9.3)	3.1 (1.5–6.1)	6.2 (2.6–14.2)
	Disagree	76.8 (73.0–80.2)	87.4 (77.9–93.2)	89.3 (77.6–95.2)	77.8 (72.1–82.7)	62.8 (53.7–71.0)	77.4 (72.7–81.5)	75.9 (68.6–82.0)
	Don’t know	9.7 (7.6–12.3)	1.7 (0.3–7.8)	0.9 (0.1–6.5)	8.7 (6.1–12.3)	19.8 (14.2–26.8)	9.3 (6.9–12.4)	11.1 (7.2–16.8)
E-cigarettes are complicated to use (*n* = 863)	Agree	22.0 (18.4–26.1)	20.9 (12.1–33.5)	8.3 (4.1–16.1)	22.0 (16.9–28.1)	30.5 (22.6–39.8)	24.3 (19.9–29.4)	14.9 (9.9–21.7)
	Neither agree nor disagree	3.3 (1.8–6.0)	1.7 (0.5–5.3)	2.6 (0.7–9.5)	3.8 (1.7–8.4)	3.3 (1.2–8.3)	2.8 (1.3–5.7)	5.1 (2.3–11.1)
	Disagree	61.3 (56.8–65.6)	76.8 (64.3–85.9)	87.1 (78.0–92.8)	63.0 (56.5–69.0)	33.3 (25.2–42.5)	60.1 (54.8–65.2)	65.5 (57.5–72.7)
	Don’t know	13.4 (11.0–16.2)	0.4 (0.1–2.9)	1.0 (0.1–7.1)	11.3 (8.3–15.2)	29.3 (23.1–36.4)	12.8 (10.2–16.1)	15.6 (11.0–21.7)
**Beliefs about harmfulness and addictiveness of vaping**								
Addictiveness of nicotine-containing ECs compared to smoking cigarettes (*n* = 857)	Less addictive	36.5 (32.3–41.0)	59.4 (47.5–70.2)	57.8 (43.8–70.7)	33.2 (27.6–39.4)	19.4 (13.4–27.1)	36.8 (31.9–41.9)	35.8 (28.2–44.2)
	Equally addictive	43.1 (38.7–47.5)	31.9 (21.9–43.9)	31.8 (20.7–45.4)	46.4 (40.3–52.6)	48.6 (40.0–57.4)	44.2 (39.1–49.5)	40.8 (32.7–49.6)
	More addictive	7.9 (5.8–10.8)	5.6 (2.3–12.9)	3.2 (1.2–8.4)	8.0 (5.1–12.4)	11.4 (6.6–18.9)	6.9 (4.6–10.2)	10.8 (6.6–17.1)
	Don’t know	12.5 (10.2–15.2)	2.0 (0.7–5.6)	6.4 (2.5–15.8)	12.5 (9.3–16.5)	19.5 (14.3–26.0)	12.1 (9.5–15.4)	13.2 (8.7–19.5)
How satisfying vaping is compared to smoking cigarettes (*n* = 671)	Less satisfying	68.3 (63.3–72.9)	53.3 (41.2–65.0)	76.9 (65.2–85.5)	70.5 (64.4–76.0)	N/A	72.1 (66.1–77.3)	58.4 (49.3–66.9)
	Equally satisfying	13.0 (9.9–16.8)	28.4 (18.0–41.6)	12.2 (6.4–22.2)	9.0 (5.9–13.3)	N/A	10.5 (7.0–15.5)	18.5 (13.1–25.4)
	More satisfying	9.2 (6.2–13.5)	15.7 (9.1–25.8)	6.9 (2.6–16.8)	8.0 (4.4–14.1)	N/A	7.7 (4.7–12.3)	13.4 (7.4–23.1)
	Don’t know	9.5 (7.3–12.4)	0.0 (No CI ^2^)	3.7 (1.4–9.5)	12.3 (9.3–16.1)	N/A	9.6 (7.0–13.1)	8.9 (5.3–14.6)
Cost of vaping or using e-cigarettes compared to smoking (*n* = 876)	Cheaper	62.4 (58.1–66.6)	94.7 (88.0–97.8)	84.9 (71.2–92.8)	63.4 (57.6–68.8)	29.4 (21.5–38.7)	63.9 (59.1–68.4)	57.7 (49.2–65.8)
	About the same	10.2 (7.8–13.2)	2.4 (0.7–8.4)	5.1 (1.5–15.6)	11.5 (8.2–15.9)	13.0 (8.0–20.5)	10.2 (7.6–13.6)	9.4 (5.4–16.0)
	More expensive	5.8 (4.1–8.2)	3.0 (0.9–9.2)	0.7 (0.2–2.7)	5.3 (3.2–8.7)	13.8 (7.7–23.5)	5.7 (3.7–8.6)	6.5 (3.3–12.4)
	Don’t know	21.5 (18.3–25.1)	0.0 (No CI ^2^)	9.4 (3.3–23.9)	19.8 (15.5–24.9)	43.3 (35.2–51.7)	20.1 (16.6–24.2)	26.5 (19.9–34.3)
Harmfulness of using e-cigarettes or vaping compared to smoking cigarettes (*n* = 861)	Less harmful	46.2 (41.8–50.6)	75.7 (63.0–85.0)	68.2 (54.2–79.6)	43.6 (37.8–49.5)	24.5 (17.8–32.8)	45.4 (40.2–50.7)	49.2 (41.7–56.8)
	No different	33.0 (29.1–37.2)	18.7 (10.1–32.0)	18.1 (10.3–30.1)	34.3 (29.1–40.0)	45.9 (36.9–55.1)	34.3 (29.5–39.4)	28.3 (21.6–36.1)
	More harmful	5.5 (3.6–8.3)	1.5 (0.3–7.6)	4.0 (1.1–13.6)	7.3 (4.6–11.5)	3.3 (1.3–8.0)	5.1 (3.1–8.3)	7.3 (3.6–14.2)
	Don’t know	15.4 (12.6–18.5)	3.1 (1.0–9.1)	9.9 (4.5–20.5)	14.7 (11.2–19.2)	25.0 (19.0–32.1)	15.3 (12.1–19.0)	15.8 (11.1–21.9)

^1^ Includes current occasional (less than monthly) users and previous users who had either tried e-cigarettes or had been a regular user in the past but were not currently using e-cigarettes; ^2^ Confidence intervals could not be calculated due to zero respondents for this response option. Percentage estimates are all marginally standardised on age–gender, ethnicity, smoking status (smoker/ex-smoker), and time in sample; the column ‘*n*’ is the number of participants in the column group with at least one valid answer to this question set, and the row ‘*n*’ is the total number of participants with valid responses to that question. N/A = not applicable.

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
