# Peer review of "Patterns of Use of Vaping Products among Smokers: Findings from the 2016–2018 International Tobacco Control (ITC) New Zealand Surveys"

_ijerph, 2020, doi:10.3390/ijerph17186629_

Round 1

Reviewer 1 Report

Patterns of use of vaping products among smokers: Findings from the 2016-2018 ITC New Zealand Surveys

The manuscript focuses on patterns of use of e-cigarettes and attitudes and beliefs among smokers and ex-smokers in New Zealand. In principle, this could be an interesting paper. However, it seems like that the authors have a holding in the e-cigarette industry. Since, although it seems as if e-cigarettes are less unhealthy than smoking conventional cigarettes, currently we do not know the exact effects of smoking e-cigarettes on health and on quitting smoking tobacco. Moreover, by using e-cigarettes, smokers still use nicotine. Therefore, I think we do not need to promote the use of e-cigarettes. I would like to suggest that the authors rewrite the paper in a more scientific way. That means that they need to explain which knowledge gap will be filled with this paper and explain what we (both scientifically and in practice) can do with these findings, without promoting the use of e-cigarettes.

  • Since most of the analyses were conducted on the Wave 2 data, it is not very clear to me why also Wave 1 data are included.
  • More information is need about the respondents of the two Waves. How many people did participate in both waves? Is it possible to conduct longitudinal analyses for the group who participated in the two waves? In other words, do we also see ‘individual’ changes over time?
  • More information is needed about marginal standardization.
  • The overall response was reasonably low. The percentages of consent to participate were much higher. What is the difference between the two percentages?
  • The authors mention that there was a substantial increase in ever use. I will believe that, looking at the percentages. It is, however, not clear to me with which statistical analysis they demonstrated this. Possibly, they can add this information. Moreover, it would be helpful for the readers when in the table with asterisks is mentioned which percentages are statistically significant different. Actually, this applies to all tables.
  • In Table 1 unweighted percentages are given. What about the weighted percentages. It seems like that in Wave 2 relatively less men and more Mãori respondents participated. This could have influenced the results and the possible differences between the two waves.
  • Including both ‘Never used vaping products’ and ‘Ever used vaping products’ (in Table 2) does not provide additional information.
  • According to the authors, (line 192/193) all indicators of use were similar between Mãori and non-Mãori and by quintiles of deprivation. Looking at Table 3, this conclusion does not seem to be correct. The ever use of Mãori is higher (AOR 1.58 (1.06-2.38)) than of non-Mãori. The same applies for deprivation 4 compared to 5. Looking at the percentages, it does not seem logical that 4 and 5 are different from each other. What is a possible explanation for this result? For the reader it would be helpful to see in the Table clearly which AORs are statistically significant. Besides, it would be helpful to put the reference category on top (now it is alternatively on top or at the bottom.
  • Table 5 is very comprehensive and therefore ‘difficult’ to read. Also in this Table it would be helpful to see at a glance which differences are statistically significant.
  • As already said, promotion of e-cigarettes should be avoided. It is not clear why we should encourage use of e-cigarettes by smokers. Therefore, a conclusion as ‘ to promote the use of vaping as a cessation tool or substitute for smoking need to focus on older age groups’ can not be drawn from the current findings in this manuscript. The same is true for promoting ‘less stringent policies for the regulation of vaping products.

Reviewer 2 Report

Well-written study exploring use of ENDS in New Zealand.  Useful information regarding reasons and barriers to use of ENDS in a population.    

Comments:

Abstract - concise and clear - no comments

Intro - Complete and detailed.

Methods - appropriate and sound.

Results - Low response rate as indicated in limitations.

Discussion - Since cost was a primary motivator for use of ENDS (i.e., ENDS are less expensive than cigarettes), and there was an increase in ENDS use from Wave 1 to 2 (from 8% to 11.4% overall) - it would be useful for audiences outside of NZ to understand the relative cost of these products (average cost of ecigarettes vs pack of cigarettes).  The reason there is a cost advantage is the high taxes of cigarettes in NZ.  Even though this might limit the generalizability of the findings, for context and period, can the authors add some indication of the NZ cigarette taxes and WHEN these high taxes went into effect? (page 15, line 65)

Flavored ENDS was also mentioned - can the authors comment on current or proposed policies regarding flavorings of tobacco products/ENDS? (around page 15, Line 54)

In discussion - although Wave 2 occurred June - Dec 2018, consider commenting as to how the change in availability of ENDS after March 2018 might impact these results of a survey done in 2019 or later. 

Round 2

Reviewer 1 Report

Since the authors did not seriously give answer to my main concerns I will not review the rest of the manuscript. I have no other choice than to reject this manuscript.

Author Response

Our response to this 'review' is attached.
